# Interrelated adolescent-level food insecurity and common mental health disorders in Harari Region, Ethiopia: A cross-sectional study

Kasiye Shiferaw[1]*, Gari Hunduma[2], Yadeta Dessie[3], Tesfaye Assebe Yadeta[2], Biftu Geda[4], Negussie Deyessa[5]

1 School of Midwifery, College of Health and Medical Sciences, Haramaya University, Harar, Ethiopia,
2 School of Nursing, College of Health and Medical Sciences, Haramaya University, Harar, Ethiopia,
3 School of Public Health, College of Health and Medical Sciences, Haramaya University, Harar, Ethiopia,
4 School of Nursing and Midwifery, College of Health and Medical Sciences, Madda Walabu University, Shashamane, Ethiopia, 5 Department of Preventive Medicine, School of Public Health, College of Health Sciences, Addis Ababa University, Addis Ababa, Ethiopia

☉ These authors contributed equally to this work.
* sifkas.gem2@gmail.com

## Abstract

### Background

Global adolescent-level food insecurity (AFI) and common mental disorders (CMD) represent a significant public health burden. While household-level food security is known to be associated with mental health disorders, there is a dearth of evidence regarding the specific association between AFI and CMDs in Ethiopia, particularly in the Harari Regional State.

### Methods

A school-based cross-sectional study was conducted among 3,326 adolescents in the Harari Regional State, Eastern Ethiopia, utilizing a multistage sampling strategy stratified by locality and school type. Data were collected using validated scales adapted from previous studies, including the HFIAS for adolescent-level food insecurity, KIDSCREEN-10 for health-related quality of life, the Rosenberg Scale for self-esteem, and the SDQ-25 for CMDs. Data were collected using a structured questionnaire translated into Amharic and Afan Oromo and pre-tested for validity. A guided self-administration approach was employed by eight trained BSc nurses and psychiatric nurses. All data collectors and supervisors underwent rigorous training to ensure data quality. Questionnaires were subjected to daily checks for completeness during fieldwork, and double data entry was utilized for final validations and accuracy. Data were analysed using STATA version 16.1. Prior to analysis, data were screened for outliers, missing values, and normality. The structural equation model (SEM) demonstrated a good fit to the data (RMSEA = 0.03, CFI = 0.90, TLI = 0.89 and SRMR = 0.05),

**Data availability statement:** All relevant data are within the manuscript and its Supporting Information files.

**Funding:** This research was funded by Haramaya university; its scientific research grant number is HURG-2020-02-01-92. The funder has no role in designing study, data collection and analysis, or manuscript writing up.

**Competing interests:** The authors have declared that no competing interests exist.

confirming the structural integrity of the model prior to the interpretation of path coefficients. We conducted SEM using maximum likelihood estimation after adjusting for relevant covariates.

## Results

The descriptive results revealed that 14.50% of adolescents experienced moderate-to-severe food insecurity, while the prevalence of CMDs was 22.93%. Several factors were significantly associated with the prevalence of CMDs. AFI was linked to a higher likelihood of CMDs ($\beta = 0.20$, $P < 0.001$), as were substance use ($\beta = 0.14$, $P < 0.001$) and frequent financial difficulties ($\beta = 0.06$, $P < 0.001$). Conversely, higher quality of life ($\beta = -0.07$, $P < 0.001$) and stronger self-esteem ($\beta = -0.06$, $P < 0.001$) were associated with a lower likelihood of CMD symptoms. Furthermore, urban residency emerged as a protective factor, showing a significant negative association with CMDs ($\beta = -0.21$, $P < 0.001$).

## Conclusion

AFI significantly predicts the occurrence of CMDs. A relationship is further driven by poor quality of life, diminished self-esteem, substance use, and financial instability. The study highlights the need for tailored interventions to tackle these functional drivers, suggesting that addressing food insecurity and its associated psychosocial factors is essential to reduce adolescents' CMD burden.

## Introduction

Ending hunger, food insecurity and all forms of malnutrition by 2030 are among sustainable development goal target 2 (SDG 2) [1]. However, around 2.3 billion people (28.0%) globally faced moderate or severe food insecurity in 2024 [2]. While global hunger showed a slight decline to 8.2% in 2024, the situation in Africa has deteriorated. Currently, approximately one in eleven people globally and one in five people in Africa suffer from chronic hunger [1,3], which may be slipping behind schedule. Driven by a combination of conflict, drought, high living cost and natural disasters, Ethiopia remains a severe food crisis zone; an estimated 15.0–15.9 million people in the country were projected to face high levels of acute food insecurity by 2026, a decrease from earlier 2025 peak [4,5].

The risk of multiple chronic health conditions; including obesity, diabetes, heart diseases, and mental health disorders are increased among population experiencing food insecurity and nutritional deficiencies [6]. Furthermore, chronic ill-health among adults may also result in household food insecurity (HFI) [7,8]. In this region, mental health association with famine and food insecurity impacts cognitive and intellectual children's development [9]. Nearly fifty percent of adolescents reported skipping breakfast meals due to food shortages, a practice associated with lower school performance [10]; various consequence of food insecurity are also identified alongside

challenges in adolescent readiness for adulthood [11]. Beyond individual well-being, food insecurity has an effect on societal problems including poverty cycles and community-level mental health [12].

Mental health problems are a heavy public health burden; nearly 14% of children and adolescents have mental health illness worldwide [13]. Even though the burden of adolescents' mental health disorder is unknown, its burden in childhood ranged from 12 to 25% in Ethiopia [14,15]. Population mental health impacted by food insecurity around the world [16] and poor mental health consistently associated with food insecurity in Africa [17]. Nutritional science evidence indicates that nutrient deficiency correlate with symptoms of depression and anxiety, potentially reflecting underlying changes in brain function; furthermore, food insecurity is identified as a significant stressor associated with mental health challenges [16].

Weiser *et al*. conceptualized the relationship of food insecurity with physical and mental health outcomes including human immune-deficiency virus (HIV) and cardiovascular health; through three primary ways: nutritional, mental and behavioral [18]. The nutritional pathway involves associations between food insecurity and malnutrition, immunological deficiencies, and poor dietary quality. The mental health pathway identifies a link between psychological stressors (such as sadness and anxiety) and an increased risk of obesity and cardiovascular diseases. Finally, the behavioral pathway highlights correlations with the consumption of calorie-dense, nutrient-poor foods, high-risk health behaviors, and challenges in medication adherence, which are associated with poorer disease management [18]. In Ethiopia, multiple micronutrients deficiencies and the 'double-burden' of malnutrition are among common problems of the adolescent population [19].

A review of the recent literature indicates that HFI and psychological distress are interrelated health issues [20,21]. Studies among children aged 5–11 years also identify an association between HFI and an increased mental health conditions diagnosis [22,23]. Similarly, evidence suggests that food-insecure mothers and women of reproductive age show a higher prevalence of stressors [24] and common mental health problems [25], respectively. A population-based study in Mexico that indicated participants across all level of food insecurity were more likely to report high depressive symptoms [26].

Global studies indicate that HFI is associated with mental health problems among adolescents [27,28] including in the USA and Canada [29]. Similarly, research in northern Ghana identified a link between depression in adolescents' girls and food insecurity [30], an observation supported by broader literature review [31]. In Ethiopia, the self-rated health status of adolescents correlates strongly with exposure to food insecurity [32], while youth from food-insecure households show a higher prevalence of CMDs [33].

In contrast, a study by Kedir *et al.* indicated that common mental disorders were not significantly associated with HFI [34]. Given the bidirectional relationship between poor mental health outcomes and food insecurity, researchers have emphasized the need for strategies to mitigate food insecurity and for health care providers to manage its presence among their clients [35]. Furthermore, a review of literature in Africa strongly suggested need of additional research within the sub-region [9]. Notably, the mediating role of adolescents' substance use including *khat (*Catha edulis) chewing, alcohol consumption, tobacco smoking, and illicit drugs use (i.e., cannabis, marijuana, shisha, cocaine or heroin), as well as quality of life and self-esteem, have not been extensively explored in prior studies.

Literature review suggest that future studies should involve adolescents directly rather than relying solely on parental reports, as they are capable and reliable participants with unique experiences [31]. Consequently, this study assesses AFI rather than HFI data to examine the relationship between this understudied factor and CMDs. To address potential psychological and physiological challenges, the conditions associated with food insecurity such as body image, psychological wellbeing and disordered eating should be managed early during adolescence [36]. These findings may inform the potential for re-introducing school-based psychological counselling and strengthening or introducing school-based meal program in Ethiopia. Therefore, this study aimed to investigate the association between AFI and CMDs among in-school adolescents in Harari Regional State, Eastern Ethiopia.

## Methods

### Study area and design

This investigation utilized a school-based cross-sectional study design. The research was conducted in the Harari Regional State of Eastern Ethiopia, located approximately 511 kilometers from the capital city. The region's economy is primarily driven by trade, with a significant portion of commercial activity centered around psychoactive substances such as coffee, tobacco, and *khat*. Parallel to these commercial trends, approximately 25% of young people chew *khat*, making it a highly prevalent practice in the community [37]. At the time of the study, 85 of the 112 schools in the region met the inclusion criteria-serving students in both primary (grades 7–8) and secondary (grades 9–12) education [38]. Data collection was carried out between November 24, 2020, and December 31, 2020.

### Population

**Population, Sample size and sampling technique.** The source population comprised all adolescents enrolled in schools within the Harari Regional State, while the study population consisted of adolescents from specifically selected schools. The sample size was determined using OpenEpi software, calculated based on a 95% confidence level, 2% precision, and an 18.5% prevalence estimate of AFI (the primary exposure variable) derived from a prior study in Ethiopia [35]. After factoring in a design effect of 2.0 and a 15% non-response rate, the final sample size for this study was 3,326 adolescents.

This study utilized a multistage sampling strategy. Of the 85 eligible schools in the region, 23 were selected using a lottery-based random sampling technique, stratified by locality (rural vs. urban) and school type (public vs. private). Subsequently, sections from each grade within the selected schools were selected at random, proportionate to the total number of students. Lastly, all students within the selected sections were invited to participate. A detailed description of this sampling procedure has been documented in prior publication [38].

### Data collection tool

The validated instruments used in this study were adapted from prior research, specifically AFI [36,39], health related quality of life [39], adolescent self-esteem [40] and adolescent's common mental health problems [41].

### Data collectors

Eight data collectors comprising psychiatry or BSc-level nurses were recruited based on their proficiency in reading and writing Amharic and Afan Oromo. Data collection was conducted over a one-month period under the supervision of three psychiatric-nurse educators who also met the necessary linguistic eligibility criteria.

### Data collection procedure

Data was collected using a guided self-administered questionnaire technique among students at their respective schools. Participants were organized into small groups of no more than 20 per session, to facilitate a suitable environment (room) for them. Participants received a comprehensive orientation regarding the study objectives and instructions for completing the questionnaire to maintain data integrity. During each session, two designated data collectors read the questions aloud, and participants selected the options that best reflected their experiences according to the established formats. The lead investigator and trained supervisors closely monitored the data collection process throughout the study period.

### Estimates of scale reliability

To determine which factor variances explained the most and to assess item loadings, scale reliability, and model fitness, we performed both exploratory and confirmatory factor analyses. Construct validity and internal consistency or scale reliability was estimated using Cronbach's alpha (α), which was calculated following confirmatory factor analysis (CFA).

## Variables and measurements

**Adolescent-level food insecurity**: AFI was assessed using a 5-item adaptation of the Household Food Insecurity Access Scale (HFIAS), based on a three-month recall period [36,39]. This version was selected for its validated utility in the Ethiopian adolescent context [32,42], as established by the Jimma Longitudinal Family Survey of Youth. It was designed to reduce respondent burden while capturing the full spectrum of food insecurity, from psychological worry to physical hunger. The five items included (1) worrying about food running out, (2) being unable to eat preferred foods, (3) having a limited variety of food, (4) eating smaller meals than needed, and (5) skipping meals due to lack of resources.

Each item was scored as 0 (No) or 1 (Yes). Following the scoring algorithm validated by Jebena *et al*. (2016) [33] on Ethiopian youth, adolescents were categorized as 'food secure' if they responded 'No' to all items. Those responding 'Yes' to items 1 or 2 (worry or quality) without reducing intake were categorized as 'mildly food insecure.' Conversely, those responding 'Yes' to any items involving reduced intake or meal frequency (items 3–5) were classified as 'moderately to severely food insecure.' This approach ensures the results reflect the specific physical and psychological thresholds of hunger relevant to this age group.

**Health related quality of life (**HRQOL**):** HRQOL was evaluated using the self-report version of the KIDSCREEN-10 Index, a validated tool for children and adolescents aged 8–18 [39]. Higher scores on this index imply a higher level of HRQOL. A comprehensive description of the scale and its application in this context is available in the study by Hunduma *et al*. [43].

**Substance use:** Using a modified questionnaire from previous studies, particularly the Global School-Based Health Survey (GSHS) [44], we asked participants one question each regarding their current smoking, *khat* chewing, and alcohol drinking habits.

**Adolescent self-esteem:** Self-esteem was measured using the Rosenberg Self-Esteem Scale, which employs a 4-point Likert scale ranging from 'strongly disagree' (1) to 'strongly agree' (4). Total scores range from 10 to 40, with higher scores indicating higher levels of self-esteem [40].

**Adolescent's CMDs:** In this study, the term 'common mental disorders' is used to categorize the specific set of psychological conditions assessed, rather than to denote their statistical commonness in the region. The Strengths and Difficulties Questionnaire (SDQ-25) was used to measure mental health status across five sub-scales: emotional symptoms, conduct problems, hyperactivity/inattention, peer relationship issues, and prosocial behavior [41]. Each item was rated on a 3-point Likert scale (0–2), and the first four sub-scale were summed to generate an overall difficulty score. This score was treated as a continuous variable, ranging from lower to higher points, where increasing values represent a higher burden of mental health difficulties.

The internal consistency of the instruments was evaluated using Cronbach's alpha. Reliability for the scales ranged from 0.62 to 0.82 (see Table 1). Specifically, KIDSCREEN-10 ($\alpha = 0.82$) demonstrated good internal consistency, while the SDQ-25 ($\alpha = 0.74$), RSES ($\alpha = 0.76$), and GSHS module ($\alpha = 0.74$) showed acceptable reliability. The 5-item AFI scale demonstrated a Cronbach's alpha of 0.62. While this falls below the conventional 0.70 threshold, it represents an acceptable level of internal consistency for an abbreviated, binary-scored instrument used in exploratory research [45]. This value likely reflects the scale's broad conceptual reach spanning from psychological worry to physical hunger and is consistent with reliability benchmarks for shortened scales in similar Ethiopian contexts.

## Covariates

**Socio-demographic status.** The respondents were questioned about their sociodemographic background, including their age, sex, place of residence, and financial difficulties. Place of residence was classified as urban or rural according to the Ethiopia Central Statistical Agency (CSA) criteria, which define urban areas as administrative capitals or localities with at least 2,000 inhabitants primarily engaged in non-agricultural activities [46], while rural areas comprise all other localities not meeting these specific criteria. Financial difficulty was measured based on the adolescent's perception of their household's economic situation. This variable was categorized into three levels: 'Never', 'Sometimes', and 'Always'

**Table 1. Psychometric properties and validation status of the data collection instruments.**

| Instrument | Construct Measured | Items | Validation Status | Study Reliability (Cronbach's alpha) |
|---|---|---|---|---|
| AFIAS | Food Insecurity | 5 | International (FAO); National (Ethiopia) | 0.62 |
| SDQ-25 | Behavioral Health | 20 | International (WHO); National (Ethiopia) | 0.74 |
| KIDSCREEN-10 | Quality of Life | 10 | International (HRQOL); National (Ethiopia) | 0.82 |
| RSES | Self-Esteem | 6 | International; Used in Ethiopian youth | 0.76 |
| GSHS Module | Substance Use | 4 | International (WHO); National (Ethiopia) | 0.74 |

Note: AFIAS: Adolescent Food Insecurity Access Scale; SDQ: Strengths and Difficulties Questionnaire; KIDSCREEN: Health-Related Quality of Life; RSES: Rosenberg Self-Esteem Scale; GSHS: Global School-Based Student Health Survey.

experiencing financial problems. This subjective classification captures the adolescent's lived experience of economic scarcity, which serves as a proximal stressor for mental health; research indicates such reports of subjective status predict adolescent well-being as effectively as, or better than, objective administrative or parental data [47].

## Data quality assurance

A structured questionnaire that was translated into Afan Oromo and Amharic was used to gather data. It was reviewed by professionals and local specialists. Under the supervision of two data collectors per session, the questionnaire was administered to teenage students in their schools after being pre-tested for validity and reliability. Before utilizing the questionnaire for the actual data collection, the necessary adjustments were made to make them clear and consistent. Five days of training on data collection were given to all supervisors and data collectors. Every day, senior investigators and experienced supervisors thoroughly monitored the data collection process. Prior to data entry, editors checked for errors and missing information. Finally, data were double entered by different data clerks to validate the entries and reduce data entry errors.

## Ethical Approval

With reference number IHRERC/149.2019, the Institutional Health Research Ethics Review Committee (IHRERC) of Haramaya University issued ethical permission. Participation was entirely voluntary; respondents were informed of their right to skip any question or withdraw at any stage without any negative consequence. All eligible participants and their guardians received clear information regarding the study's objectives, methods, risks, and benefits, with the assurance that data would be used strictly for research purposes. Written informed consent was obtained from parents or guardians for participants aged 13–17, supplemented by the adolescents' written assent. Participants aged 18 or older provided their own written informed consent. To maintain confidentiality, personal identifiers were omitted from all questionnaires. The study adhered to the ethical principles of the Declaration of Helsinki.

## Data processing and analysis

Data analysis was performed using STATA version 16.1. Prior to analysis, we examined the data for outliers, missingness, and non-normality. For each construct, item reliability was verified, and items with factor loadings below 0.4 were excluded [45,48]. We employed SEM using the Satorra-Bentler estimator to account for non-normal distribution [49]. The association between adolescent food insecurity and CMDs was evaluated through a sequential four-model approach: Model I established the independent association; Model II added quality of life; Model III incorporated substance use; and the final Model IV examined the mediating role of self-esteem while adjusting for dietary diversity and confounders. Mediation effects were tested following Baron and Kenny's methods [50].

Model fit was evaluated using Chi-square, CFI, TLI, RMSEA, and SRMR [51–53]. We applied thresholds of CFI/TLI ≥ 0.90 and RMSEA ≤ 0.05 for 'good' fit, though in our large sample (n > 3000), an RMSEA up to 0.10 was considered acceptable if supported by CFI/TLI > 0.95 and low SRMR [54,55]. Reliability was assessed using Cronbach's alpha (α > 0.70) for reflective scales. For formative indices like dietary diversity, where high inter-item correlation is not theoretically expected, we utilized Composite Reliability (CR > 0.70) and Average Variance Extracted (AVE > 0.50) to ensure convergent validity [56]. Model parsimony was improved by eliminating non-significant paths, with comparisons guided by the Akaike Information Criterion (AIC).

## Results

### Socio-demographic characteristics

A total of 3,227 participants were included in the analysis, representing a 97% response rate. Missing data across the primary variables was negligible and was handled using pairwise deletion, which did not significantly affect the statistical power or the study findings. The overall mean age was 15.47 years, with 51.75% of the sample being female. Among male adolescents, the average age was 15.91 years.

These data provide empirical support for the significant burden of food insecurity and CMDs in the study population. Specifically, 14.50% of adolescents experienced moderate-to-severe food insecurity, while 22.93% scored in the 'abnormal' range for CMDs, justifying the classification of these conditions as prevalent. The distribution and prevalence of primary outcomes are summarized in Table 2.

**Internal consistency, model fitness and factor loadings.** Exploratory factor analysis (EFA) of AFI indicated that the first factor and the total model accounted for 41% and 62% of the cumulative variance, respectively. The scale demonstrated high internal consistency (Cronbach's alpha = 0.86), with factor loadings ranging from 0.64 to 0.79. For the quality-of-life variable, the first factor and the total model explained 32% and 35% of the variance, respectively, with a Cronbach's alpha of 0.82. Finally, EFA for CMDs indicated that the first factor accounted for 31% of the variance; factor loadings for these items ranged from 0.40 to 0.55, with a reliability coefficient of 0.74. These psychometric properties are summarized in Table 3.

### Confirmatory factor analysis

We evaluated the full measurement models, and all standardized factor loadings were statistically significant at the 0.05 level. The overall model fit indices were within acceptable ranges (Chi$^2$ = 1786.25, P < 0.001; RMSEA = 0.04; pclose = 1.0; CFI = 93; TLI = 92; SRMR = 0.04; CD = 88). Additionally, the mean Variance Inflation Factor (VIF) was 1.48, indicating no significant issues with multicollinearity among the variables.

**Table 2. Descriptive summary of key study variables.**

| Variable | Category | Frequency (%) |
|---|---|---|
| Food Insecurity (HFIAS) | Moderate to severe insecurity | 468 (14.50%) |
| Mental Health (SDQ-25) | Abnormal range (CMDs) | 740 (22.93%) |
| Quality of Life (KIDSCREEN) | Low quality of life | 858 (26.58%) |
| Self-Esteem (RSES) | Low self-esteem | 344 (10.66%) |

Note: (n = 3,227). Minor variations in sub-group totals are due to occasional missing data for specific variables (<0.1%). All percentages are calculated based on the total available responses for each item.

**Table 3. Psychometric properties of study constructs: model fit indices, internal consistency, and standardized factor loadings for adolescent-level variables, Harari Region, Eastern Ethiopia, 2024.**

| Adolescent-level food insecurity | Factors loading | Goodness of fitness |
|---|---|---|
| In the last three months, how many days did you worry that you would run out of food or not have enough money to buy food? | 0.64 | • RMSEA = 0.09<br>• LR Chi² p-value = 0.000<br>• Pclose = 0.000<br>• CFI = 0.98<br>• TLI = 0.96<br>• CD = 0.87<br>• SRMR = 0.02<br>• Cronbach's alpha = 0.86<br>• CR = 0.86; AVE = 0.57 |
| In the last three months, how many days have you had to reduce the number of meals eaten in a day, because of shortages of food or money? | 0.76 | |
| In the last three months, how many days have you had to reduce the size of meals eaten in a day, because of shortages of food or money? | 0.78 | |
| In the last three months, how many days have you had to spend the whole day without eating, because of shortages of food or money? | 0.79 | |
| In the last three months, how many days have you had to ask for food or money to buy food? | 0.79 | |
| **Frequency of feeding different foods (Yes/No)** | | |
| Any food produced from grains—*injera*, *teff*, millet, sorghum, corn, rice, wheat, sugar, cookies | 0.42 | • RMSEA = 0.05<br>• LR Chi² p-value = 0.000<br>• Pclose = 0.06<br>• CFI = 0.94<br>• TLI = 0.90<br>• CD = 0.63<br>• SRMR = 0.03<br>Cronbach's alpha = 0.62<br>CR = 0.62; AVE = 0.22 |
| Any pulses (beans, lentils, peas)? | 0.55 | |
| Any nuts or seeds such as ground nuts, sesame or sunflower seeds? | 0.46 | |
| Any vegetables? | 0.52 | |
| Any fruits? | 0.41 | |
| Any dairy products—milk, cheese, yogurt (not including butter)? | 0.44 | |
| **Substance use** | | |
| How frequently do you drink alcohol (like Beer, *Teji, Tela*) in the past 12 months? | | • RMSEA = 0.08<br>• LR Chi² p-value = 0.000<br>• Pclose = 0.01<br>• CFI = 0.98<br>• TLI = 0.96<br>• CD = 0.83<br>• SRMR = 0.02<br>Cronbach's alpha = 0.74<br>CR = 0.75; AVE = 0.42 |
| How frequently do you smoke tobacco in the past 12 months? | | |
| In the last 12 months, how often were you chewing *khat*? | | |
| In the last 12 months how often were you using hard substances like cannabis, Mariwana, Heroine? | | |
| **KIDSCREEN – 10 – INDEX**<br>Thinking about the last week rate questions from never/not at all to always/extremely. | | |
| Have you felt fit and well? | 0.56 | • RMSEA = 0.10<br>• LR Chi² p-value = 0.000<br>• Pclose = 0.00<br>• CFI = 0.90<br>• TLI = 0.86<br>• CD = 0.82<br>• SRMR = 0.05<br>Cronbach's alpha = 0.82<br>CR = 0.82; AVE = 0.36 |
| Have you felt full of energy? | 0.62 | |
| Have you had enough time for yourself? | 0.60 | |
| Have you been able to do the things that you want to do in your free time? | 0.59 | |
| Have your parents treated you fairly? | 0.65 | |
| Have you had fun with your friends? | 0.60 | |
| Have you got on well at school? | 0.59 | |
| Have you been able to pay attention? | 0.59 | |
| **Rosenberg Self-Esteem Scale**<br>The questions are Likert scale ranging from strongly disagree (1) to strongly agree (5). | | |

*(Continued)*

**Table 3.** (Continued)

| Adolescent-level food insecurity | Factors loading | Goodness of fitness |
|---|---|---|
| Overall, I am satisfied with myself. | 0.65 | • RMSEA = 0.05 |
| I feel that I have several good qualities. | 0.64 | • LR Chi$^2$ p-value = 0.000 |
| I can do things as well as most other people. | 0.45 | • Pclose = 0.29 |
| I feel that I'm a person of worth, at least on an equal plane with others. | 0.54 | • CFI = 0.98 <br> • TLI = 0.96 <br> • CD = 0.78 |
| I wish I could have more respect for myself. | 0.61 | • SRMR = 0.02 |
| I take a positive attitude toward myself. | 0.66 | Cronbach's alpha = 0.76 <br> CR = 0.76; AVE = 0.36 |
| **Common Mental health problems (SDQ-25)** | | |
| I am constantly fidgeting or squirming | 0.55 | • RMSEA = 0.04 |
| I fight a lot. I can make other people do what I want | 0.50 | • LR Chi$^2$ p-value = 0.000 |
| I am often unhappy, downhearted or tearful | 0.41 | • Pclose = 0.95 |
| I am easily distracted; I find it difficult to concentrate | 0.40 | • CFI = 0.93 |
| I am nervous in new situations. I easily lose confidence | 0.51 | • TLI = 0.91 |
| I am often accused of lying or cheating | 0.43 | • CD = 0.75 |
| Other children or young people pick on me or bully me | 0.44 | • SRMR = 0.03 |
| I take things that are not mine from home, school or elsewhere | 0.47 | Cronbach's alpha = 0.74 <br> CR = 0.74; AVE = 0.21 |
| I am usually on my own. I generally play alone or keep to myself | 0.40 | |
| I get very angry and often lose my temper | 0.45 | |
| I am restless, I cannot stay still for long | 0.41 | |

AVE: Average Variance Extracted; AVE of 0.50 or higher means that the construct explains more than half of the variance of its indicators; LR Chi$^2$: Likelihood Ratio Chi square test; CFI: Comparative Fit Index; CR: Composite Reliability; CR values >0.70 indicate adequate internal consistency and construct reliability; RMSEA: Root Mean Square Error of Approximation; SRMR: Standardized Root Mean Square Residual; TLI: Tucker-Lewis's index; CD: Coefficient of Determination

## Association of AFI and CMDs

The results of the SEM that were determined using standard errors and path coefficients were shown in Table 4. To predict CMDs, the final model's fitness indices were sufficient (Chi$^2$ (682) = 3227.76, P < 0.001; RMSEA_SB = 0.03; Pclose = 1.00; CFI_SB = 0.90; TLI_SB = 0.89; CD = 0.89; SRMR = 0.05).

The analysis revealed significant associations between adolescent-level food insecurity, psychosocial factors, and mental health disorders. AFI was positively associated with a higher likelihood of CMDs ($\beta$ = 0.20, P < 0.001) and substance use ($\beta$ = 0.14, P < 0.001), while showing a strong negative association with both quality of life ($\beta$ = −0.37, P < 0.001) and self-esteem ($\beta$ = −0.41, P < 0.001).

Regarding potential protective factors, higher quality of life ($\beta$ = −0.07, P < 0.001) and elevated self-esteem ($\beta$ = −0.06, P < 0.001) were both linked to a lower likelihood of CMDs. Conversely, behavioural and economic stressors were associated with increased psychological burden; specifically, substance use ($\beta$ = 0.14, P < 0.001) and the presence of occasional ($\beta$ = 0.06, P < 0.001) or persistent financial difficulties ($\beta$ = 0.18, P < 0.001) were associated with higher CMD scores. Finally, urban residency was associated with a reduced likelihood of CMDs ($\beta$ = −0.21, P < 0.001) compared to rural residency. In the sequential SEM analysis, the direct path coefficient from AFI to CMDs decreased from $\beta$ = 0.26 in Model I to $\beta$ = 0.20 in Model IV after adjusting for quality of life and self-esteem.

**Table 4. Associations between AFI and CMDs: Unstandardized path coefficients (Harari Regional State, Eastern Ethiopia, 2024).**

| Measurement and structural model | Model I Path coefficient (SE) | Model II Path coefficient (SE) | Model III Path coefficient (SE) | Model IV Path coefficient (SE) |
|---|---|---|---|---|
| AFI (adjusted for covariates) | 0.26 (0.02)* | 0.23 (0.02)* | 0.21 (0.02)* | 0.20 (0.02)* |
| Age | 0.01 (0.004) | 0.01 (0.004) | 0.01 (0.004) | 0.01 (0.004) |
| Sex (Male) | 0.01 (0.01) | 0.01 (0.01) | 0.01 (0.01) | 0.01 (0.01) |
| Residence (Urban) | −0.24 (0.02)* | −0.22 (0.02)* | −0.21 (0.02)* | −0.21 (0.02)* |
| Financial problem (sometimes) | 0.08 (0.02)* | 0.06 (0.02)* | 0.06 (0.02)* | 0.06 (0.02)* |
| Financial problem (always) | 0.21 (0.04)* | 0.18 (0.04)* | 0.18 (0.04)* | 0.18 (0.04)* |
| Model I + Quality of Life | | −0.09 (0.01)* | −0.09 (0.01)* | −0.07 (0.01)* |
| AFI → Quality of Life | | −0.37 (0.04)* | −0.37 (0.04)* | −0.37 (0.04)* |
| Model II + Substance use | | | 0.16 (0.05)* | 0.14 (0.05)* |
| AFI → Substance use | | | 0.14 (0.01)* | 0.14 (0.01)* |
| Model III + self-esteem | | | | −0.06 (0.01)* |
| AFI → self-esteem | | | | −0.41 (0.04)* |
| **Goodness of fit** | | | | |
| RMSEA_SB | 0.04 | 0.04 | 0.03 | 0.03 |
| Pclose | 0.96 | 1.00 | 1.00 | 1.00 |
| CFI_SB | 0.90 | 0.88 | 0.87 | 0.90 |
| TLI_SB | 0.89 | 0.87 | 0.88 | 0.89 |
| CD | 0.89 | 0.88 | 0.89 | 0.89 |
| SRMR | 0.05 | 0.05 | 0.05 | 0.05 |

*Significance at p-value ≤ 0.05; AFI = Adolescent-level Food Insecurity; CFI_SB = Comparative Fit Index Satorra-Bentler; RMSEA_SB = Root Mean Square Error of Approximation Satorra-Bentler; SRMR = Standardized Root Mean Square Residual; TLI_SB = Tucker-Lewis's index Satorra-Bentler; CD = Coefficient of Determination

## Discussion

This study aimed to examine the association between CMDs and AFI in Harari Region of Ethiopia. According to the current study, adolescents in Harari Region, Ethiopia, who experience food insecurity exhibit higher prevalence of CMDs. Similarly, factors such as lower quality of life, diminished self-esteem, rural residency, substance use, and persistent financial difficulties were significantly associated with an increased likelihood of reporting CMDs.

Food insecure adolescents had a higher likelihood of having common mental disorders. Similarly, review of recent literature revealed that HFI and psychological distress are interrelated health issues [20,21]. Studies among children aged 5–11 years also indicated HFI is associated with an increased mental health conditions diagnosis [22,23]. Similarly, evidence revealed food insecure mothers and women of reproductive age experience stressors [24] and common mental health problems [25], respectively. A population-based study in Mexico indicated participants with all types of food insecurity were more likely to have high depressive symptoms [26].

Studies across the globe indicated that HFI is associated with overall mental health problems in adolescents [27,28] in USA and Canada [29], depression in adolescents' girls in northern Ghana [30] and review also echoed this evidence [31]. Studies from 95 countries also echoed adolescents' food insecurity associated with poorer mental health [57]. In Ethiopia youth whose households are food insecure are associated with CMDs [33]. A link between food insecurity and general health was highlighted by the study by Vuong *et al*. 2022, which discovered independent associations between lower physical and mental component scores and HFI [58].

As a result, having insufficient access to food is theoretically linked to increased stress and worry, which are associated with poorer mental health outcomes. From a nutritional perspective, inadequate intake of protein and micronutrients,

such as iron, may be associated with alterations in neurotransmitter production, potentially contributing to changes in mood, cognition, and energy levels. Such deficiencies are often correlated with symptoms including mood swings, cognitive decline, and irritability [59–61]. This complex interrelationship between mental health and food security necessitates comprehensive research and interventions that address both domains. In alignment with Sustainable Development Goals 2 and 3, this study explores the associations between AFI and mental health to support global wellbeing.

Adolescents experiencing food insecurity exhibited lower quality of life; conversely, those with higher quality of life scores showed a lower likelihood of common mental health disorders. The growing amount of research indicates that the relationship between hunger, mental health, and social functioning significantly correlates with quality of life in low- and middle-income countries [62]. Despite a scarcity of evidence regarding these associations among adolescents, both food insecurity and poor mental health are associated with reduced quality of life among people living with HIV in Ethiopia [63].

Similarly, food insecure adolescents were more likely to report lower self-esteem, suggesting that the impact of food insecurity transcends physical hunger. In Ethiopia, the self-rated health status of adolescents is strongly correlated with exposure to food insecurity [32] and youth whose households exhibit a higher prevalence CMDs [33]. According to a 2019 study by Godrich *et al.*, children from food-insecure homes demonstrated poorer self-esteem and fewer healthy decision-making behaviors compared to their food-secure peers [64]. In the social environment of school-going adolescents, the inability to meet basic food needs often leads to social comparison and internalized stigma. This erosion of self-worth acts as a psychological stressor; adolescents may perceive their food status as a personal or familial failure, which directly fuels the development of CMDs through a diminished sense of agency and social belonging.

Urban residency was associated with a lower likelihood of CMDs, although this relationship is not consistent across all literature. Some study indicates that living in urban areas is correlated with a lower prevalence of certain mental health conditions [65–67], however, other suggest that urban environments are linked to a higher risk of anxiety and depression, potentially mediated by factors such as social isolation and increased stress exposure. Although the underlying mechanisms remain unclear, living in urban has been associated with a higher reporting of specific mental health disorders [68].

Urban environments correlate with higher rates of anxiety, depression, and some psychotic diseases, according to recent evaluations of the urban environment on mental health. Concentrated poverty, inadequate social capital, social segregation, and other social and environmental adversities that are more common in urban areas are associated with an increased risk for mental disorders [69]. Furthermore, epidemiological evidence shows a statistical relationship between urban residency and an elevated frequency of schizophrenia, as mental health conditions are often reported at higher rates in urban regions compared to rural ones. While the precise nature of these connections remains under study, social isolation, prejudice, and neighborhood poverty are significant risk markers identified alongside the mental health burden in these areas [70].

The findings do not imply that urbanicity is an etiologic factor for mental diseases, even though the prevalence of disorders is somewhat higher in small urban and semi-rural areas [71]. There is limited evidence of a relationship between the urban public realm and adolescent mental health and wellbeing, which some claim highlights the difficulty of studying the intricate connections between settings and health [72]. There are various ways to work toward the objective of comprehending how urban settings relate to mental health issues. Economic factors (unemployment, socioeconomic position), social conditions (support from social networks), and environmental exposures (toxins, air pollution, noise, and light) can all be measured to see how they correlate with CMDs [72].

Food insecure adolescents showed a higher statistical likelihood of substance use; similarly, substance use was linked to a higher frequency of CMDs. Review in Africa indicated, substance abuse and food insecurity were significantly correlated, with food-insecure adolescents showing a higher pooled prevalence ratio of substance use [73]. Among students, food insecurity and drug use frequently co-occur [74]. Regardless of the environment, many conditions associated with adolescent-level food insecurity, especially those involving behavior, psychological development, mental health, and academic performance are identified alongside challenges in preparing for adulthood [10]. The results showed a strong

association between alcohol consumption and a few external variables, including food insecurity, stress, peer pressure, and attitudes toward alcohol [75].

Similarly, adolescents who experienced financial problems showed a higher statistical likelihood of CMDs. Numerous research associating young people's mental health problems with socioeconomic disadvantages supports this observation [76,77]. The results of the review demonstrated a consistent connection between childhood and adolescent mental health issues and socioeconomic hardship [78]. A child's poorer mental health status correlates with family income, though this association is less pronounced when factoring in other variables, such as maternal psychological distress [79].

A critical finding is the significant attenuation of the direct path coefficient between AFI and CMDs from Model I to Model IV. This reduction indicates that the relationship between hunger and mental health is not purely direct but is partially mediated by psychosocial factors, specifically self-esteem and health-related quality of life. This 'attenuation' suggests that while food insecurity is the root cause, its impact on mental health is channeled through the deterioration of an adolescent's psychological and social well-being.

Given that our findings highlight self-esteem and quality of life as key mediators, interventions must be school-centric. We recommend the implementation of targeted school feeding programs to provide immediate physical relief and reduce the social stigma of hunger. Furthermore, school-based psychological support groups focusing on resilience and self-worth should be integrated into existing health services. Such dual-pronged approaches addressing both nutritional deficits and psychosocial erosion are likely to be more effective compared to broad policy changes in improving adolescent mental health.

## Strength and limitation of the study

We have recruited enough participants to draw meaningful conclusions and generalize about the source population. We have adopted appropriate tools to measure outcomes and explanatory variables. Appropriate model of analysis was used and researcher ensured conclusion match the data.

However, since we tested a psycho-social and a behavioral pathway, and broader socioeconomic stressors, these findings represent statistical associations rather than confirmed causal mechanisms. It is possible that the observed relationships are bidirectional or influenced by unmeasured confounding factors. Future longitudinal research is required to confirm these pathways and determine the direction of causality over time. Another notable limitation is the exclusion of out-of-school adolescents, which may limit the generalizability of the findings to the broader adolescent population, particularly those in more vulnerable socioeconomic circumstances.

While our measurement models demonstrated strong internal consistency and incremental fit, some indices, specifically the RMSEA for KIDSCREEN-10 and AFI (0.09–0.10), fell into the borderline range. Additionally, several constructs exhibited low AVE, particularly in the SDQ-25 (0.21) and dietary diversity (0.22) scales. These values likely reflect the inherent challenge of capturing complex, multi-domain behaviours in a cross-cultural setting where items (e.g., distinct food groups or diverse mental health symptoms) are not theoretically expected to be highly redundant.

However, because these constructs achieved acceptable Composite Reliability (CR > 0.60) and excellent residuals (SRMR < 0.05), they remain valid representations of the study population's behaviours. Nonetheless, these borderline fit indices and low shared variance suggest that the underlying factor structures should be interpreted with caution and further validated in future research involving diverse adolescent populations.

Further, due to reliance on adolescent self-reports to measure financial difficulty, students may lack a comprehensive understanding of objective household economic indicators, such as total monthly income or specific asset ownership. Hence, future studies could benefit from triangulating these reports with parental data or household asset indices.

## Conclusion

There is a strong association between CMDs and adolescent-level food insecurity. Common mental health disorders are also highly correlated with substance use, low self-esteem, low quality of life, as well as financial difficulties. Interventions

for CMDs among adolescents should address the circumstances of food insecurity, substance use, self-esteem, quality of life, and financial difficulties that adolescents encounter daily. Given the cross-sectional nature of this research, these findings identify important correlations rather than causal relationships; however, they suggest that adolescent mental health strategies may benefit from integrated approaches that address food insecurity and broader socioeconomic stressors. Future longitudinal research involving both in-school and out-of-school populations is required to confirm these pathways and further validate the measurement models across diverse adolescent settings.

## Supporting information

**S1 File. Data.**
(XLS)

## Author contributions

**Conceptualization:** Gari Hunduma.

**Data curation:** Kasiye Shiferaw, Gari Hunduma.

**Formal analysis:** Kasiye Shiferaw, Gari Hunduma.

**Investigation:** Gari Hunduma.

**Methodology:** Kasiye Shiferaw, Gari Hunduma, Yadeta Dessie, Tesfaye Assebe Yadeta, Biftu Geda, Negussie Deyessa.

**Software:** Gari Hunduma.

**Supervision:** Yadeta Dessie, Tesfaye Assebe Yadeta, Biftu Geda, Negussie Deyessa.

**Validation:** Kasiye Shiferaw, Gari Hunduma, Yadeta Dessie, Tesfaye Assebe Yadeta, Biftu Geda, Negussie Deyessa.

**Visualization:** Gari Hunduma.

**Writing – original draft:** Kasiye Shiferaw, Gari Hunduma, Yadeta Dessie, Tesfaye Assebe Yadeta, Biftu Geda, Negussie Deyessa.

**Writing – review & editing:** Kasiye Shiferaw, Gari Hunduma, Yadeta Dessie, Tesfaye Assebe Yadeta, Biftu Geda, Negussie Deyessa.

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
