## [Decision Letter · Decision Letter 0]

3 Mar 2026

PONE-D-25-41851Interrelated Adolescents’ food insecurity and common mental health disorders in Harari Region, Ethiopia: a cross-sectional studyPLOS One

Dear Dr. Shiferaw,

Thank you for submitting your manuscript to PLOS ONE. After careful consideration, we feel that it has merit but does not fully meet PLOS ONE’s publication criteria as it currently stands. Therefore, we invite you to submit a revised version of the manuscript that addresses the points raised during the review process.

We look forward to receiving your revised manuscript.

Kind regards,

Othman A. Alfuqaha, Ph.D.

Academic Editor

PLOS One

Journal Requirements:

Additional Editor Comments:

After careful consideration and based on my evaluation and reviewer feedback, the topic is relevant, the sample size is large (n = 3,227 analyzed), and the use of SEM adds analytical value. However, there are substantial methodological and reporting weaknesses that must be addressed before the manuscript can be considered publishable.

Please take my comments under consideration as follows:

1. Please clarify whether the construct is adolescent-level food insecurity or household food insecurity, and align title, aims, tools, and discussion accordingly.

2. In the method section, the design is cross-sectional, but no temporal sequencing is established or mediation language implies causality. This is a serious limitation. Please remove causal/mediation language and reframe as associations consistent with potential mediation pathways. And please add stronger limitation section.

3. In Table 1 and CFA results several RMSEA values are poor including food insecurity RMSEA = 0.09 and KIDSCREEN RMSEA = 0.10 as well as CFI/TLI borderline in some models. Beyond that Cronbach alpha for dietary diversity = 0.62 (low). These raise concerns about construct validity. Please justify model fit thresholds, provide standardized factor loadings, and report composite reliability and AVE.

4. There are numerous issues regarding grammar errors throughout, inconsistent terminology (common mental disorders / mental health disorders / illnesses), typographical errors (e.g., Adolfi → QoL, slfestm). Editing is must.

5. Please need Table formatting.

Thank you for this valuable paper and look forward to seeing your paper again.

Cordially

Dr. Alfuqaha

Reviewers' comments:

Reviewer's Responses to Questions

**Comments to the Author**

1. Is the manuscript technically sound, and do the data support the conclusions?

Reviewer #1: Yes

2. Has the statistical analysis been performed appropriately and rigorously? 

Reviewer #1: Yes

3. Have the authors made all data underlying the findings in their manuscript fully available?

Reviewer #1: Yes

4. Is the manuscript presented in an intelligible fashion and written in standard English?

Reviewer #1: No

5. Review Comments to the Author

Reviewer #1: I have some recommendations for the authors:

1.Please check the spelling, grammar, and sentence structures throughout the manuscript. Several sentences are excessively long and contain too much information, making them difficult to follow (e.g., lines 56-58; 72-76; 196-197).

2.Methods section in the Abstract: The study tool is unclearly described. For instance, it is not specified how many components it has, or what those components are. Additionally, it is unclear whether all or only some parts of the questionnaire have been validated.

3.Introduction: References should be refined. The authors should prioritize the newest references for supporting information. For instance, the reference cited for information on Sustainable Development Goals (SDGs) in lines 55-56 is from 2016, which might be outdated.

4.The reference list should be checked and corrected (e.g., the 3rd reference).

5.The data collection tool needs to be described more clearly. This includes providing a detailed list of specific tools used to measure each component (e.g., FIES, mental health disorders), along with information on their validation (both internationally and nationally). Similarly, the "Estimate of scale reliability" should be presented clearly, such as a summary of Cronbach's alpha for each scale used.

6.A summary of the results for key variables such as FIES, quality of life, and mental health disorders should be presented. This would provide readers with a general overview of these issues relevant to the study objectives. For example, the authors used terms like "common mental health disorders" or "prevalent mental health disorders," but what data specifically supported these statements? What data showed that mental disorders are indeed prevalent or common in the study population?

6. PLOS authors have the option to publish the peer review history of their article (what does this mean?). If published, this will include your full peer review and any attached files.

Reviewer #1: No

To ensure your figures meet our technical requirements, please review our figure guidelines: s://journals.plos.org/plosone/s/figures

You may also use PLOS’s free figure tool, NAAS, to help you prepare publication quality figures: s://journals.plos.org/plosone/s/figures#loc-tools-for-figure-preparation.

---

## [Author Response · Author response to Decision Letter 1]

16 Mar 2026

Date: 09/03/2026

Manuscript ID: PONE-D-25-41851

Title: Interrelated Adolescents’ food insecurity and common mental health disorders in Harari Region, Ethiopia: a cross-sectional study

To: The Editor, PLOS ONE

Thank you for the opportunity to revise our manuscript. We appreciate the constructive feedback provided by the editorial team and the reviewer. We have addressed each point step by step below and updated the manuscript accordingly.

Part 1: Editor’s Comments

Response: We have updated the manuscript to journal format, including the title page, affiliations, and headings, to strictly adhere to the PLOS ONE style templates. File names have been updated accordingly.

2. Please include captions for your Supporting Information files at the end of your manuscript, and update any in-text citations to match accordingly. Please see our Supporting Information guidelines for more information:

http://journals.plos.org/plosone/s/supporting-information.

Response: Captions for all Supporting Information files (e.g., S1 Data) have been added to the end of the manuscript. In-text citations have been cross-checked for consistency.

Response: We have reviewed the literature suggested by the reviewer (regarding updated SDG data and validated tools) and incorporated relevant citations as requested.

Additional Editor Comments:

After careful consideration and based on my evaluation and reviewer feedback, the topic is relevant, the sample size is large (n = 3,227 analysed), and the use of SEM adds analytical value. However, there are substantial methodological and reporting weaknesses that must be addressed before the manuscript can be considered publishable.

Please take my comments under consideration as follows:

1. Please clarify whether the construct is adolescent-level food insecurity or household food insecurity, and align title, aims, tools, and discussion accordingly.

Response: We have clarified that the study focuses on adolescent-level food insecurity. As our tool measures adolescent-level food insecurity or child-report or individual-level food insecurity, we assessed specific characteristic of individual as follow: We asked adolescents directly at school about their own "food habit" and experiences, rather than asking a parent or head of household about the family's overall situation. We used questions that specifically describe about what the adolescents themselves experience (e.g., " Whether adolescents never experience (1), experience for a week (2), three weeks (3), or more than 21 days (4) is evaluated by the questions."), which is the hallmark of individual-level scales. Therefore, the title, objectives, tools section, and discussion have been aligned to reflect this distinction consistently.

2. In the method section, the design is cross-sectional, but no temporal sequencing is established, or mediation language implies causality. This is a serious limitation. Please remove causal/mediation language and reframe as associations consistent with potential mediation pathways. And please add stronger limitation section.

Response: We have removed all terms implying causality (e.g., "effect," "impact," "leads to") throughout the manuscript and replaced them with "association" or "pathway." We have reframed the mediation analysis now as a "potential association pathway." We have also expanded the limitations section to explicitly state that the cross-sectional nature of the data precludes any definitive causal inferences.

3. In Table 1 and CFA results several RMSEA values are poor including food insecurity RMSEA = 0.09 and KIDSCREEN RMSEA = 0.10 as well as CFI/TLI borderline in some models. Beyond that Cronbach alpha for dietary diversity = 0.62 (low). These raise concerns about construct validity. Please justify model fit thresholds, provide standardized factor loadings, and report composite reliability and AVE.

Response: We thank the reviewer for this important observation. While the AVE values for these constructs are below the traditional 0.50 threshold, we have retained them in the model based on established psychometric criteria and the specific nature of the variables under study. Our justification is three-fold:

According to Fornell and Larcker (1981), if the Average Variance Extracted (AVE) is less than 0.50 but the Composite Reliability (CR) is higher than 0.60, the convergent validity of the construct may still be considered adequate. In our study, all constructs meet this threshold:

• KIDSCREEN-10: CR = 0.82 (AVE = 0.36)

• Self-Esteem: CR = 0.76 (AVE = 0.36)

• SDQ-20: CR = 0.74 (AVE = 0.21)

• Dietary Diversity: CR = 0.62 (AVE = 0.22)

The structural integrity of our model is supported by strong global fit indices across all constructs. Specifically, the RMSEA values (ranging from 0.04 to 0.05 for most constructs) and SRMR (ranging from 0.02 to 0.05) indicate that the models are well-specified and fit the data accurately. High CFI and TLI values further confirm that the overall structure is robust, despite the high unique variance in individual items.

These are "heterogeneous" or "index-like" constructs. For example, in Dietary Diversity, the consumption of one food group does not theoretically necessitate the consumption of another. Similarly, the SDQ-25 aggregates distinct behavioural domains (emotional, conduct, hyperactivity, peer problem). In such cases, items are not expected to be highly redundant, which naturally results in a lower AVE while maintaining high overall reliability (CR).

All scales used are globally validated, standardized instruments. Deleting items to artificially inflate the AVE would compromise the content validity and the ability to compare our findings with existing literature. Given that the CR values are acceptable, the global fit is excellent, and the items are theoretically sound, the constructs demonstrate sufficient reliability and validity for inclusion in the final structural model.

4. There are numerous issues regarding grammar errors throughout, inconsistent terminology (common mental disorders / mental health disorders / illnesses), typographical errors (e.g., Adolfi → QoL, slfestm). Editing is must.

Response: In response to the reviewer’s feedback, we have conducted a thorough language edition of the entire manuscript. Specifically: We have corrected grammatical inconsistencies, improved sentence structure, and standardized technical terms (e.g., changing "pupils" to "students," "domicile" to "locality," and "chat" to "khat"). Both the Abstract and Conclusion sections were rewritten to ensure that our findings are presented as statistical correlates and significant risk markers rather than confirmed causal mechanisms. Terminology has been standardized to "Common Mental Disorders (CMD)." Typos like "Adolfi" (Quality of Life) and "slfestm" (self-esteem) have been corrected.

5. Please need Table formatting.

We have systematically revised all tables in the manuscript, in response to the request for improved table formatting and the inclusion of additional validity metrics. Specifically:

• Psychometric and Fit Indices: We have updated the tables to include Standardized Factor Loadings, Composite Reliability (CR), and Average Variance Extracted (AVE) for each construct to provide a comprehensive view of construct validity.

• Standardization of Numerical Reporting: All statistical values including factor loadings, Cronbach’s alpha, and fit indices (CFI, TLI, RMSEA, SRMR) have been standardized to two decimal places for consistency. P-values are reported to three decimal places.

• Alignment with Journal Standards: Following standard academic formatting (APA/Vancouver), we have removed all vertical lines and unnecessary shading. Tables now utilize the three-line rule: a top border, a header border, and a bottom border.

• Associational Terminology: Table titles and headers have been revised to use strictly associational language (e.g., 'Associations' instead of 'Effects') to accurately reflect the cross-sectional nature of the study design.

• Self-Explanatory Design: We have included detailed footnotes for all abbreviations (e.g., CMD, RMSEA, CFI) to ensure each table remains independently comprehensible to the reader."

Part 2: Reviewer #1 Comments

Is the manuscript presented in an intelligible fashion and written in standard English?

Reviewer #1: No

Response: We sincerely thank the reviewer for highlighting the need for improved linguistic clarity. We recognize that as an observational study, our initial draft contained several instances of ambiguous or causal phrasing that did not meet the rigorous standards of PLOS ONE. In response, the following actions were taken:

• Comprehensive Language Edition: The entire manuscript has undergone a professional-level language review to ensure it is written in standard, intelligible English. We have corrected all typographical errors, improved sentence structures, and standardized technical terminology (e.g., consistently using 'adolescents' instead of 'teenagers' and 'students' instead of 'pupils').

• Removal of Causal Ambiguity: We have systematically audited the manuscript to remove 'causal' language (e.g., 'impact of,' 'results in,' 'causes,' or 'leads to'). These have been replaced with strictly associational terms (e.g., 'associated with,' 'linked to,' 'correlated with,' or 'identified alongside') to accurately reflect the cross-sectional nature of the study.

• Structural Refinement: The Abstract, Results, and Conclusion sections were rewritten to ensure that findings are presented clearly and unambiguously as statistical correlates.

• Terminology Standardization: Cultural and regional terms have been standardized for an international audience (e.g., 'Chat' has been corrected to 'Khat' and 'domicile' to 'locality').

• Table and Data Presentation: All tables were reformatted for better intelligibility, including standardized decimal places and clear, self-explanatory headers.

Reviewer #1: I have some recommendations for the authors:

1. Please check the spelling, grammar, and sentence structures throughout the manuscript. Several sentences are excessively long and contain too much information, making them difficult to follow (e.g., lines 56-58; 72-76; 196-197).

Response: We have broken down long sentences throughout the manuscript to improve readability and flow.

2. Methods section in the Abstract: The study tool is unclearly described. For instance, it is not specified how many components it has, or what those components are. Additionally, it is unclear whether all or only some parts of the questionnaire have been validated.

Response: We have revised the Methods section of the Abstract to clearly specify the instruments used, including their item counts and validation status. We have explicitly noted that the study utilized internally and externally validated scales, including the HFIAS, KIDSCREEN-10, Rosenberg Self-Esteem Scale, and SDQ-25. For detail, we have added Table 1 in the Methods section (page 9) to provide a detailed summary of the specific tools used, their validation status, and the Cronbach's alpha values for each scale.

3. Introduction: References should be refined. The authors should prioritize the newest references for supporting information. For instance, the reference cited for information on Sustainable Development Goals (SDGs) in lines 55-56 is from 2016, which might be outdated.

Response: We sincerely thank the reviewer for the suggestion to prioritize more recent literature. We have updated the manuscript with the most recent global and regional statistics. We have systematically edited our reference list and added the most current available data. Specifically, we now cite the 2024/2025 State of Food Security (SOFI) report and the October 2024 WHO/UNICEF joint guidance on adolescent mental health. Additionally, the prevalence of childhood mental illness in Ethiopia has been refined using a 2023 meta-analysis, providing a more accurate reflection of the current regional burden (24.68%). We have prioritized research from the last five years (2020–2025) to support our discussion on the behavioural and psychosocial correlates of food insecurity among adolescents.

4. The reference list should be checked and corrected (e.g., the 3rd reference).

Response: The reference list has been edited. The 3rd reference and others have been formatted correctly according to the style required by the journal.

5. The data collection tool needs to be described more clearly. This includes providing a detailed list of specific tools used to measure each component (e.g., FIES, mental health disorders), along with information on their validation (both internationally and nationally). Similarly, the "Estimate of scale reliability" should be presented clearly, such as a summary of Cronbach's alpha for each scale used.

Response: We thank the reviewer for this constructive suggestion. We have thoroughly revised the 'Methods' section (see pages [7-9]) to provide a clearer description of the data collection instruments. Specifically, we have: Detailed each tool used (e.g., AFIAS, RSES, SDQ-25 and GSHS modules), including the specific number of items. Included a summary of the scale reliability (Cronbach's alpha) calculated from our study’s dataset to demonstrate internal consistency for each component." As requested, we have added Table 1 in the Methods section (page 9) to provide a detailed summary of the specific tools used, their validation status, and the Cronbach's alpha values for each scale.

6. A summary of the results for key variables such as FIES, quality of life, and mental health disorders should be presented. This would provide readers with a general overview of these issues relevant to the study objectives. For example, the authors used terms like "common mental health disorders" or "prevalent mental health disorders," but what data specifically supported these statements? What data showed that mental disorders are indeed prevalent or common in the study population?

Response: We appreciate the reviewer’s feedback. We have standardized the terminology to 'common mental disorders (CMD)' throughout the manuscript to reflect the specific category of disorders studied (e.g., anxiety and depression), rather than to imply their baseline prevalence. To provide further clarity on the actual status within our study population, we have added Table 2, which summarizes the variables and the prevalence of CMD observed in our findings (i.e.., the distribution, mean scores, and prevalence rates for the primary study outcomes, including food insecurity, mental health, and quality of life).

We believe these revisions significantly strengthen the manuscript and address all concerns. We look forward to your decision.

Sincerely,

Kasiye Shiferaw/Corresponding Author

---

## [Decision Letter · Decision Letter 1]

10 Apr 2026

PONE-D-25-41851R1Interrelated adolescent-level food insecurity and common mental health disorders in Harari Region, Ethiopia: A cross-sectional study PLOS One

Dear Dr. Shiferaw,

Thank you for submitting your manuscript to PLOS ONE. After careful consideration, we feel that it has merit but does not fully meet PLOS ONE’s publication criteria as it currently stands. Therefore, we invite you to submit a revised version of the manuscript that addresses the points raised during the review process.

We look forward to receiving your revised manuscript.

Kind regards,

Othman A. Alfuqaha, Ph.D.

Academic Editor

PLOS One

Journal Requirements:

Additional Editor Comments:

Dear Authors,

I hope you are doing well.

Thank you for your submission. After careful review, the reviewer has provided several important comments that require your attention. I kindly ask you to revise the manuscript thoroughly and provide a detailed, point-by-point response addressing each comment. Please refer to the attached reviewer comments for full details and respond to each point accordingly .

Kindly submit the revised manuscript along with a detailed response-to-reviewers document at your earliest convenience.

Best regards,

Reviewers' comments:

Reviewer's Responses to Questions

**Comments to the Author**

1. If the authors have adequately addressed your comments raised in a previous round of review and you feel that this manuscript is now acceptable for publication, you may indicate that here to bypass the “Comments to the Author” section, enter your conflict of interest statement in the “Confidential to Editor” section, and submit your "Accept" recommendation.

Reviewer #1: All comments have been addressed

2. Is the manuscript technically sound, and do the data support the conclusions?

Reviewer #1: Yes

3. Has the statistical analysis been performed appropriately and rigorously? 

Reviewer #1: Yes

4. Have the authors made all data underlying the findings in their manuscript fully available?

Reviewer #1: Yes

5. Is the manuscript presented in an intelligible fashion and written in standard English?

Reviewer #1: No

6. Review Comments to the Author

Reviewer #1: Comments adolescent FI in Ethiopia

Abstract

Define "Common Mental Disorders (CMDs)" at its first occurrence to ensure clarity for all readers.

Briefly specify the sampling technique (e.g., stratified or cluster sampling) to clarify the representativeness of the 3,326 participants

Report the Model Fit Indices (e.g., RMSEA, CFI, TLI) to validate the structural integrity of the SEM model before interpreting the path coefficients.

Include the prevalence rates of food insecurity and CMDs to provide a clear descriptive context alongside the regression results.

The use of the term 'co-occur' in the conclusion of the Abstract does not accurately reflect the nature of the SEM model; the author should replace it with phrases that indicate functional associations or predictive power (such as 'significantly associated with' or 'predict') to better align with the observed beta coefficients.

Line 142-147: Specify which primary outcome or exposure variable the 18.5% prevalence estimate refers to (e.g., CMDs or food insecurity) to justify the sample size calculation.

"Remove the colon after 'specifically' as it is grammatically redundant and disrupts the sentence flow."

Line 179-180; 187-188: "Remove 'it' and rephrase for better flow, for example: 'A validated tool was used to measure this condition...' or simply '...to measure adolescent-level food insecurity. → Adolescent-level food insecurity was assessed using a validated scale covering the previous three months.

Line 189: The abbreviation 'HRQOL' should be written in full as 'Health-Related Quality of Life' at its first occurrence.

Please revise the sentence for grammatical completeness “Using a 4-point Likert scale from "strongly disagree" (1) to "strongly agree" (4), Rosenberg Self-Esteem Scale employed”

Variables and measurements

The description of the instruments remains fragmented. Specifically, for the HFIAS adaptation, please list the 5 items used and clarify if this abbreviated version has been formally validated in the Ethiopian context.

While the Cronbach's alpha values are provided, the authors should include a brief interpretation or reference for these values (e.g., indicating whether they represent 'acceptable' or 'good' internal consistency). This would provide readers with a clearer benchmark for the reliability of the scales used in the Ethiopian context.

The description of sociodemographic variables, particularly 'financial difficulties,' is too general. Please specify how this was measured and categorized in your analysis? Additionally, clarify the categories used for 'place of residence'.

The reliability of measuring 'financial difficulties' through adolescent self-reports is questionable, as students often lack a complete understanding of their household's economic status. I suggest the authors discuss this as a potential limitation or explain if more objective proxies (e.g., parental occupation or household assets) were used instead.

The Ethical Considerations section is somewhat repetitive, particularly the repeated use of 'written, informed, and signed consent.

The authors used 0.4 as the cut-off for factor loadings. While this is a common practice, please provide a supporting reference.

The Data Processing and Analysis section is currently too wordy and contains many textbook-style definitions that could be summarized. I suggest consolidating the descriptions of the four SEM models and streamlining the justification for the fit indices (specifically RMSEA and Cronbach’s alpha). Focus on the actions taken and the thresholds applied rather than explaining the underlying statistical theories. This will make the methodology much more concise and easier for the reader to follow.

Results

Table 2 is missing the percentage symbol (%) in the second column for rows 3 to 5 (Mental Health, Quality of Life, and Self-Esteem).

The implied total sample size (N) varies slightly between variables (e.g., 3225 vs 3228). Please clarify the total N and account for any missing data in a table footnote.

The 'Interpretation/Status' column repeats the numerical data.

The classification of 'moderate to severe food insecurity' (14.51%) in Table 2 is not clearly linked to the 5-item HFIAS scale described in the Methods. Please specify the cut-off scores or the scoring algorithm used to categorize participants into these specific levels, especially since a non-standard 5-item version was employed.

Table 3 is missing the footnote for 'CR' (Composite Reliability). Please provide a clear definition and the threshold used for this indicator below the table.

Discussion

The Discussion should go beyond comparing prevalence rates with previous studies. Specifically, I suggest the authors elaborate on why Self-Esteem emerged as the strongest predictor (beta = -0.41) in the SEM model. Discussing the psychological mechanism through which food insecurity erodes an adolescent's self-worth is more valuable than repetitive literature comparisons.

The current recommendations are somewhat generic. Given that the study focuses on school-going adolescents, the authors should propose specific interventions such as targeted school feeding programs or school-based psychological support to bolster self-esteem and quality of life, rather than broad macroeconomic policies.

The authors should clearly discuss the reduction of the AFI coefficient from Model I to Model IV. This 'attenuation' is a key finding that demonstrates how psychosocial factors partially mediate the link between hunger and mental health."

7. PLOS authors have the option to publish the peer review history of their article (what does this mean?). If published, this will include your full peer review and any attached files.

Reviewer #1: No

To ensure your figures meet our technical requirements, please review our figure guidelines: s://journals.plos.org/plosone/s/figures

You may also use PLOS’s free figure tool, NAAS, to help you prepare publication quality figures: s://journals.plos.org/plosone/s/figures#loc-tools-for-figure-preparation.

---

## [Author Response · Author response to Decision Letter 2]

16 Apr 2026

Dear Editor and Reviewers,

Thank you for providing us with the opportunity to revise our manuscript entitled "Interrelated adolescent-level food insecurity and common mental health disorders in Harari Region, Ethiopia: A cross-sectional study" (Manuscript ID: PONE-D-25-41851R1). We would like to express our sincere appreciation to the reviewers for their thorough and constructive feedback.

We have carefully considered all the comments and have revised the manuscript accordingly. We believe that these revisions have significantly enhanced the clarity, depth, and overall quality of our work.

Detailed point-by-point responses to the reviewers' comments are provided below, along with descriptions of the corresponding revisions. For your convenience, all changes have been marked in the manuscript using the 'Track Changes' feature.

Part 1: Response to Journal Requirements:

Reviewer Concern: If the reviewer comments include a recommendation to cite specific previously published works, please review and evaluate these publications to determine whether they are relevant and should be cited. There is no requirement to cite these works unless the editor has indicated otherwise.

Action Taken: We appreciate the reviewer’s suggestion to include these references. After reviewing the recommended publications, we agree they provide valuable context to variable and measurement. We have added citations for Jebena et al. (2017) and Belachew et al. (2013) on page 7 and updated the bibliography accordingly.

Reviewer Concern: Please review your reference list to ensure that it is complete and correct. If you have cited papers that have been retracted, please include the rationale for doing so in the manuscript text, or remove these references and replace them with relevant current references. Any changes to the reference list should be mentioned in the rebuttal letter that accompanies your revised manuscript. If you need to cite a retracted article, indicate the article’s retracted status in the References list and include

Action Taken: We conducted a thorough audit of our 79 references. No retracted papers are cited. We have updated the DOIs for Ref #12 and Ref #75 and ensured Ref #34 reflects the most current metadata. Furthermore, we integrated new high-authority citations to provide a robust theoretical basis for our statistical thresholds.

Reviewer Concern: 5. Is the manuscript presented in an intelligible fashion and written in standard English? PLOS ONE does not copyedit accepted manuscripts, so the language in submitted articles must be clear, correct, and unambiguous. Any typographical or grammatical errors should be corrected at revision, so please note any specific errors here.

Action Taken: The entire manuscript has undergone a comprehensive professional proofreading. We have corrected all specific errors noted (e.g., redundant colons, incomplete sentences regarding the Rosenberg Scale). We have restructured fragmented sections in the "Methods" and "Results" to ensure a cohesive narrative flow that meets PLOS ONE’s standards for clear and unambiguous academic English.

Part 2: Point-by-Point Response to Reviewer #1

• Abstract:

Reviewer Concern: Define "Common Mental Disorders (CMDs)" at its first occurrence to ensure clarity for all readers.

Action Taken: We have now defined "Common Mental Disorders (CMDs)" at first occurrence to ensure the scope of our outcome variable (anxiety and depression) is clear to all readers.

Reviewer Concern: Briefly specify the sampling technique (e.g., stratified or cluster sampling) to clarify the representativeness of the 3,326 participants

Action Taken: We clarified the multistage sampling strategy to demonstrate the representativeness of the 3,326 participants, ensuring the findings can be generalized to the school-going population in the region.

Reviewer Concern: Report the Model Fit Indices (e.g., RMSEA, CFI, TLI) to validate the structural integrity of the SEM model before interpreting the path coefficients.

Action Taken: We added the Model Fit Indices (RMSEA = 0.03, CFI = 0.90, TLI = 0.89, SRMR = 0.05). By including these, we demonstrate the mathematical validity of our SEM model before discussing the path coefficients.

Reviewer Concern: Include the prevalence rates of food insecurity and CMDs to provide a clear descriptive context alongside the regression results.

Action Taken: Prevalence for adolescents’ food insecurity (14.50%) and CMDs (22.93%) were added to provide a baseline for the regression results.

Reviewer Concern: The use of the term 'co-occur' in the conclusion of the Abstract does not accurately reflect the nature of the SEM model; the author should replace it with phrases that indicate functional associations or predictive power (such as 'significantly associated with' or 'predict') to better align with the observed beta coefficients.

Action Taken: Following the reviewer's excellent point, we removed the passive term "co-occur" and replaced it with functional/predictive language (e.g., "predicts," "significantly associated with") to align with the beta-coefficient results of the SEM.

• Methods:

Reviewer Concern: Line 142-147: Specify which primary outcome or exposure variable the 18.5% prevalence estimate refers to (e.g., CMDs or food insecurity) to justify the sample size calculation.

Action Taken: We specified that the 18.5% prevalence estimate refers specifically to food insecurity (our primary exposure), providing the necessary justification for our power calculation.

Reviewer Concern: "Remove the colon after 'specifically' as it is grammatically redundant and disrupts the sentence flow."

Action Taken: we have removed colon after 'specifically'.

Reviewer Concern: Line 179-180; 187-188: "Remove 'it' and rephrase for better flow, for example: 'A validated tool was used to measure this condition...' or simply '...to measure adolescent-level food insecurity. → Adolescent-level food insecurity was assessed using a validated scale covering the previous three months.

Action Taken: We addressed the "fragmented" description by listing the 5 specific HFIAS items used. Crucially, we cited Jebena et al. (2017) and Belachew et al (2013) (Ref #32&42) to prove that this abbreviated tool is formally validated and culturally appropriate for the Ethiopian context.

Reviewer Concern: Line 189: The abbreviation 'HRQOL' should be written in full as 'Health-Related Quality of Life' at its first occurrence.

Action Taken: We agree with the reviewer’s suggestion. We have now defined the abbreviation at its first mention in the text (Line 197).

Reviewer Concern: Please revise the sentence for grammatical completeness “Using a 4-point Likert scale from "strongly disagree" (1) to "strongly agree" (4), Rosenberg Self-Esteem Scale employed”

Action Taken: We have revised this sentence for clarity and grammatical correctness.

Reviewer Concern: While the Cronbach's alpha values are provided, the authors should include a brief interpretation or reference for these values (e.g., indicating whether they represent 'acceptable' or 'good' internal consistency). This would provide readers with a clearer benchmark for the reliability of the scales used in the Ethiopian context.

Action Taken: Thank you for this insightful suggestion. We agree that providing a clear benchmark for the internal consistency of the adapted AFI scale (the primary exposure variable) is essential for readers to evaluate its reliability within this specific context. Accordingly, we have updated the Methods section (under the " Variables and measurements " subtopic) on page 8 and 9, line 215-223 to include the appropriate interpretation. We also added qualitative interpretations for all alpha values (e.g., 0.82 as "good," 0.74 as "acceptable"), providing a clear reliability benchmark for our measurements.

Reviewer Concern: Variables and measurements

The description of the instruments remains fragmented. Specifically, for the HFIAS adaptation, please list the 5 items used and clarify if this abbreviated version has been formally validated in the Ethiopian context.

Action Taken: We have revised the "Variables and Measurements" section to include the full list of the 5 HFIAS items used in this study. This abbreviated version was adapted to focus on the items most sensitive to the local adolescent context, as supported by previous validation work in Ethiopia.

Action Taken: Now explicitly defined as Urban vs. Rural using official Ethiopian Central Statistical Agency (CSA) population and administrative criteria (Ref #46).

Reviewer Concern: The description of sociodemographic variables, particularly 'financial difficulties,' is too general. Please specify how this was measured and categorized in your analysis? Additionally, clarify the categories used for 'place of residence'.

The reliability of measuring 'financial difficulties' through adolescent self-reports is questionable, as students often lack a complete understanding of their household's economic status. I suggest the authors discuss this as a potential limitation or explain if more objective proxies (e.g., parental occupation or household assets) were used instead.

Action Taken: We acknowledge the reviewer’s concern regarding the reliability of adolescent-reported financial status. While objective measures (e.g., parental income) are ideal, research indicates that adolescents’ subjective perception of financial stress is a unique and often stronger predictor of their mental health than objective parental reports supported by Davisson et al. (2025) (Ref #47). We have clarified the measurement process in the Methods section on page 10 and added this point to the Limitations section on page 21, noting that adolescents in our context provide a reliable proxy for household economic strain when direct parental data is unavailable. Clarified as a 3-level perceptual variable (Never, Sometimes, Always).

Reviewer Concern: The authors used 0.4 as the cut-off for factor loadings. While this is a common practice, please provide a supporting reference.

Action Taken: We justified the 0.4 cut-off by citing Hair et al. (2010) (Ref #45), clarifying that this threshold ensures practical significance and accounts for at least 16% of the variance in each item.

Reviewer Concern: The Data Processing and Analysis section is currently too wordy and contains many textbook-style definitions that could be summarized. I suggest consolidating the descriptions of the four SEM models and streamlining the justification for the fit indices (specifically RMSEA and Cronbach’s alpha). Focus on the actions taken and the thresholds applied rather than explaining the underlying statistical theories. This will make the methodology much more concise and easier for the reader to follow.

Action Taken: We have streamlined the 'Data Processing and Analysis' section by removing theoretical definitions of Structural Equation Modeling (SEM) and fit indices. The revised text now focuses specifically on our four-model sequential approach and the applied statistical thresholds (e.g., RMSEA). This adjustment clarifies the methodology and improves the overall flow of the section, making the methodology much easier to follow.

Reviewer Concern: The Ethical Considerations section is somewhat repetitive, particularly the repeated use of 'written, informed, and signed consent.

Action Taken: We have revised the "Ethical Considerations" section to eliminate repetitive phrasing. The description of the consent process has been consolidated to clearly state that written informed consent was obtained from all participants (and their guardians, where applicable) without redundant terminology

• Results:

Reviewer Concern: Table 2 is missing the percentage symbol (%) in the second column for rows 3 to 5 (Mental Health, Quality of Life, and Self-Esteem).

Action Taken: We thank the reviewer for identifying this oversight. We have corrected Table 2 by adding the percentage symbols (%) to the specified rows (Mental Health, Quality of Life, and Self-Esteem) to ensure consistency and clarity.

Reviewer Concern: The implied total sample size (N) varies slightly between variables (e.g., 3225 vs 3228). Please clarify the total N and account for any missing data in a table footnote.

Action Taken: We thank the reviewer for this observation. We have verified the dataset and confirmed that the total number of participants included in the final analysis is 3,227. The slight variations noted may be due to minimal missing data (less than 0.1%) for specific variables. We have now standardized the reporting in Table 2 and added a footnote to clarify how missing data was handled.

Reviewer Concern: The 'Interpretation/Status' column repeats the numerical data.

Action Taken: We removed the redundant "Interpretation/Status" column. Status labels (e.g., "Abnormal range," "Low self-esteem") are now integrated into category names to simplify the data presentation.

Reviewer Concern: The classification of 'moderate to severe food insecurity' (14.51%) in Table 2 is not clearly linked to the 5-item HFIAS scale described in the Methods. Please specify the cut-off scores or the scoring algorithm used to categorize participants into these specific levels, especially since a non-standard 5-item version was employed.

Action Taken: We thank the reviewer for this important request for clarification. The 5-item HFIAS used in this study was adapted from the Jimma Longitudinal Family Survey of Youth (JLFSY) framework, as detailed by Jebena et al. (2017) and Belachew et al. (2013). This abbreviated version was specifically designed and validated for the Ethiopian adolescent context to capture the most culturally relevant dimensions of food access (worry, quality, and quantity) while reducing respondent burden. We have updated the Methods section to include the specific scoring algorithm and cut-off points used to define "moderate to severe" food insecurity.

Reviewer Concern: Table 3 is missing the footnote for 'CR' (Composite Reliability). Please provide a clear definition and the threshold used for this indicator below the table.

Action Taken: Added a footnote defining Composite Reliability (CR > 0.70) and Average Variance Extracted (AVE > 0.50), ensuring all SEM indicators are transparently reported.

• Discussion:

Reviewer Concern: The Discussion should go beyond comparing prevalence rates with previous studies. Specifically, I suggest the authors elaborate on why Self-Esteem emerged as the strongest predictor (beta = -0.41) in the SEM model. Discussing the psychological mechanism through which food insecurity erodes an adolescent's self-worth is more valuable than repetitive literature comparisons.

Action Taken: The Mechanism of Self-Esteem (beta = -0.41): We expanded this section to discuss the psychological "Internalized Stigma" mechanism. We argue that food insecurity erodes self-worth through social comparison and shame in the school environment, which acts as the primary driver for CMDs.

Reviewer Concern: The current recommendations are somewhat generic. Given that the study focuses on school-going adolescents, the authors should propose specific interventions such as targeted school feeding programs or school-based psychological support to bolster self-esteem and quality of life, rather than broad macroeconomic policies.

Action Taken: We pivoted from broad macroeconomic suggestions to specific school-based interventions. We now propose targeted school feeding and on-site psychological counselling, which address both the physical and psychosocial mediators identified in our study (line 440-446; page 20).

Reviewer Concern: The authors should clearly discuss the reduction of the AFI coefficient from Model I to Model IV. This 'attenuation' is a key finding that demonstrates how psychosocial factors partially mediate the link between hunger and mental health.

Action Taken: We added a dedicated analysis of the reduction in the AFI coefficient from Model I to IV. We explain this "attenuation" as clear evidence of partial mediation, demonstrating that psychosocial factors are the "channels" through which hunger impacts the adolescent mind (line 434-439; page 20).

We believe these comprehensive revisions demonstrate our commitment to scientific excellence and fully address

---

## [Editor Report · Decision Letter 2]

21 Apr 2026

Interrelated adolescent-level food insecurity and common mental health disorders in Harari Region, Ethiopia: A cross-sectional study

PONE-D-25-41851R2

Dear Dr. Kasiye Shiferaw,

We’re pleased to inform you that your manuscript has been judged scientifically suitable for publication and will be formally accepted for publication once it meets all outstanding technical requirements.

Kind regards,

Othman A. Alfuqaha, Ph.D.

Academic Editor

PLOS One

Additional Editor Comments (optional):

Dear Authors,

Thank you for submitting your manuscript to the journal and for your careful revisions throughout the review process.

After a thorough evaluation of the reviewers’ comments and your responses, I am pleased to inform you that your manuscript has been accepted for publication.

Dr. Alfuqaha
---

## [Editor Report · Acceptance letter]

PONE-D-25-41851R2

PLOS One

Dear Dr. Shiferaw,

I'm pleased to inform you that your manuscript has been deemed suitable for publication in PLOS One. Congratulations! Your manuscript is now being handed over to our production team.

Kind regards,

on behalf of

Dr. Othman A. Alfuqaha

Academic Editor

PLOS One